# Extracellular Vesicles and Cx43-Gap Junction Channels Are the Main Routes for Mitochondrial Transfer from Ultra-Purified Mesenchymal Stem Cells, RECs

**DOI:** 10.3390/ijms241210294

**Published:** 2023-06-18

**Authors:** Jiahao Yang, Lu Liu, Yasuaki Oda, Keisuke Wada, Mako Ago, Shinichiro Matsuda, Miho Hattori, Tsukimi Goto, Shuichi Ishibashi, Yuki Kawashima-Sonoyama, Yumi Matsuzaki, Takeshi Taketani

**Affiliations:** 1Department of Pediatrics, Faculty of Medicine, Shimane University, 89-1 Enya-cho, Izumo 693-8501, Japan; jiahao@med.shimane-u.ac.jp (J.Y.); m209432@med.shimane-u.ac.jp (L.L.); y-oda@med.shimane-u.ac.jp (Y.O.); keiwada@med.shimane-u.ac.jp (K.W.); ago-mako@med.shimane-u.ac.jp (M.A.); mhatt001@med.shimane-u.ac.jp (M.H.); t.goto@med.shimane-u.ac.jp (T.G.); yuki.kawashima@med.shimane-u.ac.jp (Y.K.-S.); 2Department of Medical Oncology, Shimane University Hospital, 89-1 Enya-cho, Izumo 693-8501, Japan; smatsuda@med.shimane-u.ac.jp; 3Department of Digestive and General Surgery, Faculty of Medicine, Shimane University, 89-1 Enya-cho, Izumo 693-8501, Japan; shuichi466@gmail.com; 4Department of Life Science, Faculty of Medicine, Shimane University, 89-1 Enya-cho, Izumo 693-8501, Japan; ymatsuzak@gmail.com

**Keywords:** rapidly expanding clones (RECs), mesenchymal stem cells (MSCs), extracellular vesicles (Evs), Cx43-gap junction channels (Cx43-GJCs), mitochondrial transfer

## Abstract

Mitochondria are essential organelles for maintaining intracellular homeostasis. Their dysfunction can directly or indirectly affect cell functioning and is linked to multiple diseases. Donation of exogenous mitochondria is potentially a viable therapeutic strategy. For this, selecting appropriate donors of exogenous mitochondria is critical. We previously demonstrated that ultra-purified bone marrow-derived mesenchymal stem cells (RECs) have better stem cell properties and homogeneity than conventionally cultured bone marrow-derived mesenchymal stem cells. Here, we explored the effect of contact and noncontact systems on three possible mitochondrial transfer mechanisms involving tunneling nanotubes, connexin 43 (Cx43)-mediated gap junction channels (GJCs), and extracellular vesicles (Evs). We show that Evs and Cx43-GJCs provide the main mechanism for mitochondrial transfer from RECs. Through these two critical mitochondrial transfer pathways, RECs could transfer a greater number of mitochondria into mitochondria-deficient (ρ^0^) cells and could significantly restore mitochondrial functional parameters. Furthermore, we analyzed the effect of exosomes (EXO) on the rate of mitochondrial transfer from RECs and recovery of mitochondrial function. REC-derived EXO appeared to promote mitochondrial transfer and slightly improve the recovery of mtDNA content and oxidative phosphorylation in ρ^0^ cells. Thus, ultrapure, homogenous, and safe stem cell RECs could provide a potential therapeutic tool for diseases associated with mitochondrial dysfunction.

## 1. Introduction

Mitochondria are double membrane-bound organelles with an outer membrane, an inner membrane (cristae), a matrix space surrounded by cristae, and an intermembrane space between the inner and outer membranes [1]. They are unique in having their own genetic material called mitochondrial DNA (mtDNA), which encodes rRNA, tRNA, and proteins essential for oxidative phosphorylation (OXPHOS) and a self-repair mechanism [2,3]. However, the relatively inefficient mechanism of mtDNA repair makes it susceptible to damage and mutations [4]. Mutations or defects in mtDNA are the most common cause of mitochondrial dysfunction and are closely associated with various diseases and cancers [5,6,7]. These disorders are challenging from clinical and therapeutic perspectives because they can affect almost any organ, and their complexity is exacerbated by the involvement of both nuclear and mitochondrial genes encoding mitochondrial proteins. Thus, searching for a possible strategy to treat or alleviate mitochondrial dysfunction is imperative.

Therapies based on bone marrow-derived mesenchymal stem cells (BMSCs) have produced beneficial effects in a wide range of cell and animal disease models. MSCs can donate mitochondria to recipient cells through multiple pathways [8,9,10]. Transfer of functional mitochondria from MSCs has also been demonstrated in different tissues and organs; these transferred mitochondria can promote cytoprotection and partial restoration of mitochondrial function in a variety of target cells, including cardiomyocytes, endothelial cells, and alveolar epithelial cells [11,12,13].

With regard to the route of targeted transfer of mitochondria from MSCs, the role of intercellular junctional structures on mitochondrial transfer based on cell-to-cell contact has been reported [14,15]. The typical pathways for the transfer of mitochondria involve tunneling nanotubes (TNTs) and connexin 43 (Cx43)-mediated gap junction channels (GJCs). TNTs are intercellular linkage channels that facilitate cell-to-cell communication over long distances. They were found to play an essential role in MSC-mediated mitochondrial transfer as early as 2006 [12]. Several studies have revealed the inhibition of MSC-mediated mitochondrial transfer upon inhibiting actin polymerization using cytochalasin D, which in turn inhibits TNT formation [13,14,16]. Another essential transmembrane protein, Cx43, which mediates GJCs, was found to participate in mitochondrial transfer from BMSCs [17] and to restore alveolar bioenergetics [18]. Other studies based on extracellular vesicles (Evs) include exosomes (EXO) (30–150 nm) and microvesicles (MVs > 150 nm)), which are essential intercellular communication vehicles released by MSCs; these studies suggest that they can target depolarized mitochondria to the plasma membrane and, thereby, regulate intracellular oxidative stress [19]. In addition, EXO that do not carry mitochondria contain nucleic acids, including mtDNA, that could be transferred to the lung in vivo over long periods [20]. These findings suggest a possible role of Evs in the transfer of mitochondria. However, the role of Evs in MSC biology remains largely unresolved. Owing to a variety of mitochondrial metastasis pathways in MSCs, the mechanisms of metastasis are intricate and complex. Therefore, deciphering the optimal pathway for the transfer of mitochondria is an important topic of research.

Although the therapeutic value of the exogenous mitochondrial replacement method (MSCs rescuing recipient cells by donating mitochondria) has been shown, the degree of proliferation and differentiation of MSCs isolated from human bone marrow using traditional methods is not consistent, and the obtained cell population is highly heterogeneous [21,22,23]. This affects experimental precision and reproducibility, and the results obtained are often contradictory. To overcome these problems, we proposed the use of rapidly expanded clones (RECs) in a previous report [22]. RECs, which are ultra-purified MSCs screened using two antibodies, LNGFR (CD271) and THY-1 (CD90), are genetically stable, ultrapure, and highly homogeneous [24,25,26]. Therefore, we hypothesized that RECs should be valuable donors of mitochondria to recipient cells.

To test the abovementioned hypothesis, we established and used contact and noncontact coculture systems to evaluate the differences in the efficiency of mitochondrial transfer and the recovery of mitochondrial function of recipient cells after transfer through different mitochondrial transfer pathways of RECs compared with conventional BMSCs. We unraveled the main mitochondrial transfer pathway of RECs and compared the efficiency of mitochondrial transfer using RECs and BMSCs. In addition, we found that REC-derived EXO facilitate the mitochondrial transfer from RECs. These findings provide valuable insights that should help in devising a new strategy for the treatment of diseases related to mitochondrial dysfunction.

## 2. Results

### 2.1. Mitochondrial Transfer to ρ^0^ Cells from RECs in a Direct-Contact System

To determine whether RECs and BMSCs transferred mitochondria into mitochondria-deficient A549 cells (ρ^0^ cells), we stained mitochondria in RECs and BMSCs using the fluorescent probe, MitoTracker Green FM (green fluorescence) and the cell membrane of ρ^0^ cells using DiD (red fluorescence). The stained cells were cocultured in a direct-contact system (24 h). Using microscopic analysis, we found that most of the cocultured cells exhibited distinct fluorescence signals. Nonetheless, REC- or BMSC-derived mitochondria exhibiting green fluorescence could be seen in a higher number of red ρ^0^ cells (white arrows), and ρ^0^ cells received multiple mitochondrial clusters (red arrows). This demonstrates the ability of RECs and BMSCs to donate mitochondria to ρ^0^ cells (Figure 1A,B). We further examined the efficiency of mitochondrial transfer at different times of coculture (0, 4, 12, 24, and 48 h). Flow cytometry analysis showed the presence of a double-positive population of DiD/MitoTracker-stained cells in the coculture (Figure 1C). Transfer of mitochondria from RECs to ρ^0^ cells was more efficient after 4 h of coculture than that from BMSCs (Figure 1D). Furthermore, the ratio of mitochondrial transfer from RECs or BMSCs to ρ^0^ cells at 24 and 48 h was similar, indicating that the efficiency of mitochondrial transfer peaked at 24 h of coculture. Notably, mitochondrial uptake was enhanced in ρ^0^ A549 cells compared with that in A549 cells, indicating that mitochondrial defects increased the uptake of mitochondria from RECs and BMSCs (Figure 1D).

### 2.2. Mitochondrial Transfer via TNTs

Recent studies have shown that the F-actin-regulated formation of TNTs offers an effective bridge for the transfer of mitochondria between healthy stem cells and injured somatic cells, and restores mitochondrial functions in injured cells [12,13,14]. In vitro studies were used to investigate the involvement of TNTs in the mitochondrial transfer of RECs and BMSCs. MitoTracker Red-stained RECs or BMSCs were cocultured with ρ^0^ cells for 24 h. Phalloidin staining (green fluorescent phalloidin conjugate that selectively binds to F-actins) revealed that TNTs connected RECs and BMSCs to ρ^0^ cells, and mitochondria could be efficiently transferred from RECs and BMSCs to ρ^0^ cells through this pathway (Figure 2A,B). This result suggests that TNTs might be involved in one of the effective pathways for mitochondrial transfer from RECs and BMSCs. In addition, to verify whether ρ^0^ cells can receive leaked mitochondria in the absence of TNT formation, we treated cells with cytochalasin D in a contact system, which effectively inhibited the formation of TNTs without affecting endocytosis [27,28]. We could barely observe the formation of TNTs by RECs and BMSCs (Figure 2C), and the rate of mitochondrial transfer from RECs and BMSCs to ρ^0^ cells was significantly decreased. However, even with the addition of cytochalasin D, which inhibited the formation of TNTs, RECs could still donate more mitochondria to ρ^0^ cells compared with BMSCs (Figure 2D). We also found that RECs formed significantly fewer TNTs than BMSCs did (Figure 2E). These findings suggest that the pathway of mitochondrial transfer through TNTs may not be the most efficient for RECs.

### 2.3. Transfer of REC- and BMSC-Derived Mitochondria to ρ^0^ Cells via Cx43-Regulated GJCs

In previous studies using an in vivo model of lung injury, BM-MSCs were shown to participate in mitochondrial transfer to alveolar epithelial cells in a calcium ion-dependent manner through the generation of connexin-43-containing gap junction channels (Cx43-GJCs) [18]. We verified the role played by Cx43 during the mitochondrial transfer from RECs/BMSCs using in vitro experiments. MitoTracker Red-stained RECs or BMSCs were cocultured with Hoechst33342 (blue)-labeled ρ^0^ cells for 24 h. Subsequent Cx43 protein staining (green) in cocultured cells revealed high expression of Cx43 protein between RECs/BMSCs and ρ^0^ cells, suggesting the possibility of the formation of gap junction plaque (Figure 3A, white arrows). To verify whether ρ^0^ cells can receive leaked mitochondria without the formation of GJCs, we added GAP26 (a peptide corresponding to residues 63–75 of connexin 43), a gap junction blocker, to the coculture system. It effectively blocked the formation of gap junction plaque (Figure 3B) and significantly decreased the rate of mitochondrial transfer from both RECs and BMSCs to ρ^0^ cells (Figure 3E). However, ρ^0^ cells still received some mitochondria, suggesting that there are other pathways involved in mitochondrial transfer. In addition, the immunoblotting analysis showed high expression of Cx43 in ρ^0^ cells cocultured with RECs (Figure 3C,D). These results suggest a potential role of Cx43-regulated GJCs in mitochondrial transfer from RECs.

### 2.4. REC-Derived Mitochondrial Transfer to ρ^0^ Cells and mtDNA Recovery in the Noncontact System

To investigate possible ways of mitochondrial transfer other than direct contact, we established a noncontact system (Figure 4B). We seeded MitoTracker Green-stained RECs and BMSCs in 3 μm cell culture inserts (cells could not pass through the insert, but the passage of mitochondria, MVs, and EXO was possible) and DiD-stained ρ^0^ cells in the bottom layer of the culture plate and cocultured them for 24 h. We found REC-/BMSC- derived mitochondria in ρ^0^ cells (Figure 4A(I,II)). To verify that ρ^0^ cells received mitochondria donated by RECs or BMSCs and that results were not false positive due to restaining of ρ^0^ cells with excess dye, we added Dynasore to the coculture system. Dynasore acts as a potent inhibitor of the endocytic pathway and rapidly blocks the formation of coated vesicles, preventing mitochondria from being delivered and received in the form of MVs. Interestingly, we could hardly find mitochondria derived from RECs or BMSCs in ρ^0^ cells (Figure 4A(III,IV)). RECs could transfer more mitochondria to ρ^0^ cells than BMSCs could in the absence of Dynasore in the noncontact system (Figure 4C). Notably, in the contact-free REC system, mitochondrial transfer was completely blocked after Dynasore treatment, but not in BMSCs (Figure 4C). Therefore, we examined the differences in the rates of mitochondrial transfer from RECs and BMSCs at different Dynasore concentrations. At 20 μM concentration, there was no significant difference in the transfer efficiency between RECs and BMSCs. At 80 μM, the mitochondrial transfer of RECs almost disappeared, whereas BMSCs still retained more of the transferred mitochondria. This reflects the sensitivity of RECs to endocytosis inhibitors (Figure 4F). To test the effect of TNT on the 3 μm noncontact system, we added an inhibitor of actin polymerization (Cytochalasin D) and found that mitochondrial transmission was unaffected (Figure 4D). In the noncontact system, we found that the 0.4 μm cell culture insert almost blocked most of the mitochondrial transfer, whereas the 3 μm cell culture insert allowed mitochondria to pass through and more mitochondria were transferred from RECs to ρ^0^ cells in the 3 μm cell culture insert (Figure 4E). We also examined the recovery of mtDNA in ρ^0^ cells in the noncontact system. The successful establishment of the ρ^0^ cell line (no mtDNA expression, but normal nDNA expression) was confirmed using PCR (Figure 4G). Real-time PCR analysis showed that the mtDNA (COX1 and HVR2) content of ρ^0^ cells was significantly higher after the coculture with RECs or BMSCs, and RECs supplemented more mtDNA to ρ^0^ cells than BMSCs did, with no significant difference in the nDNA content between the groups (Figure 4H), which is consistent with previous reports [29,30]. These results indicate the advantage of RECs in restoring mtDNA. Moreover, using transmission electron microscopy (TEM), we observed that the mitochondria of RECs had normal morphology and regularly ordered cristae (Figure 4I(I)). The mitochondria of ρ^0^ cells were significantly swollen and had irregular, interspersed, and disordered cristae (Figure 4I(II)). When ρ^0^ cells were cocultured with RECs (Figure 4I(III)) or BMSCs (Figure 4I(IV)) in a noncontact system, we found a lowering in the degree of mitochondrial swelling as well as in cristae irregularity, dispersion, and disorder in ρ^0^ cells. The mitochondrial swelling in ρ^0^ cells was significantly reduced after coculture with RECs, compared with that achieved after coculture with BMSCs (Figure 4J). Notably, in ρ^0^ cells cocultured with RECs or BMSCs, we found the presence of mitochondria in MVs (Figure 4I, blue arrows) and structures similar to EXO (Figure 4I, red arrows). Based on these observations, we speculated that EVs might have a role in the transfer of mitochondria.

### 2.5. Characterization of Exosomes

Exosomes were isolated from the culture supernatants of RECs and BMSCs. The TEM results showed that REC-exosomes (R-EXO) and BMSC-exosomes (M-EXO) are round or oval and vary in size (Figure 5A). The diameters of REC-EXO and MSC-EXO were in the 30–150 nm range, typical of EXO (Figure 5B). R-EXO had a smaller particle size compared with M-EXO (Figure 5C). The isolated EXO were characterized with EXO-specific markers, TSG101, CD9, and CD81. WB analysis showed that REC-EXO and MSCs-EXO isolated from RECs and BMSCs, respectively, expressed TSG101, CD9, and CD81, whereas, upon the addition of the EXO biogenesis/release inhibitor, GW4869, no EXO markers were detected (Figure 5D). The above results indicate that EXO of REC and BMSC origin were successfully obtained.

### 2.6. REC-EXO Contributes to the Donation of Mitochondria by RECs and BMSCs to ρ^0^ Cells

To test the effect of EXO on the efficiency of mitochondrial donation from RECs and BMSCs to ρ^0^ cells, we cocultured cells in the noncontact system, which avoids the effects of information exchange from direct contact and indirectly inhibits the formation of TNTs and GJCs. We added EXO derived from RECs or BMSCs at the bottom of the cell culture plate (Figure 6A), in the presence or absence of GW4869 (EXO biogenesis/release inhibitor). Prior to this, we compared the effect of R-EXO and M-EXO at different concentrations on the rates of mitochondrial transfer from RECs and BMSCs. It was found that the rates of mitochondrial transfer from RECs and BMSCs were significantly higher at 40 µg (~3 × 10^11^ exosomes containing 40 µg protein) for R-EXO (Figure 6B) and M-EXO (Figure 6C). Therefore, we chose to add 40 µg of R-EXO and M-EXO. Flow cytometry showed a significant decrease in a mitochondrial donation to ρ^0^ cells by either RECs or BMSCs after the addition of GW4869. The mitochondrial transfer rate decreased significantly more for RECs than for BMSCs (Figure 6D). These results indicate the high sensitivity of RECs to GW4869. Moreover, we found that after adding more REC-EXO or BMSCs-EXO, both RECs and BMSCs donated more mitochondria to ρ^0^ cells (Figure 6D). The possible reason for this is that exosomes act as a mediator of intercellular communication, carrying abundant innate cargoes, including lipids, proteins, and nucleic acids that facilitate the exchange of substances between cells. In addition, Manickam et al. reported that exosomes naturally bind mitochondria or mitochondrial fragments during their biogenesis [31], which highlights the possibility that functional mitochondria are bound for transport into ρ^0^ cells. However, no significant difference was found when we compared the effect of different sources of exosomes on the rate of mitochondrial transfer from RECs or MSCs.

### 2.7. REC-EXO Contributes to the Restoration of Mitochondrial Function in ρ^0^ Cells

The noncontact system in which RECs can better restore mitochondrial function in ρ^0^ cells is shown in Appendix A. We further investigated the effect of different sources of EXO on mitochondrial function in ρ^0^ cells that received mitochondria donated by RECs. Using the JC-1 assay, we found that the mitochondrial membrane potential (MMP) of ρ^0^ cells was significantly higher after the addition of REC-EXO or MSC-EXO compared with that in the REC treatment group alone (Figure 7A,B), but there was no significant difference in intracellular mitochondrial ROS levels (Figure 7C,D). We also examined the difference in the mDNA content between groups after the addition of EXO and found that *COX1* and *HVR2* mtDNA content was significantly increased in ρ^0^ cells after the addition of REC-EXO compared with that in the REC-alone treatment group, with no significant difference in nDNA (*CH1*) (Figure 7E,F). These results show the advantage of REC-EXO in restoring the mitochondrial DNA content of ρ^0^ cells. We examined the expression of Cx43 after adding only REC-EXO or MSC-EXO to ρ^0^ cells. REC-EXO could better activate the expression of Cx43 than M-EXO (Figure 7G,H). To investigate the effect of EXO on the bioenergetics of ρ^0^ cells, we examined the oxygen consumption rate (OCR) using the Seahorse technique. The bioenergetic profiles of different groups are shown in Figure 8A. The basal OCR of ρ^0^ cells was significantly lower than that of the remaining groups, and after the addition of REC-EXO to ρ^0^ cells cocultured with RECs, the basal OCR was significantly restored, and the basal OCR of ρ^0^ cells after the addition of REC-EXO was superior to that achieved with RECs alone (Figure 8B). The addition of oligomycin (blocking oxidative phosphorylation and electron transport chain) and assessment of ATP synthesis by measuring OCR revealed that RECs or RECs combined with EXO increased ATP synthesis in ρ^0^ cells (Figure 8C). FCCP was added and maximum mitochondrial respiration was determined based on OCR. We found that RECs combined with REC-EXO significantly regulated the maximal respiration of ρ^0^ cells (Figure 8D). The addition of the compound I inhibitor, rotenone (AA/Rot), and spare capacity assay by measuring OCR revealed that RECs or RECs combined with EXO increased the spare capacity of ρ^0^ cells (Figure 8E), indicating a possible role of REC-EXO in restoring mitochondrial bioenergy in ρ^0^ cells.

## 3. Discussion

The applicability of MSCs as donors of exogenous mitochondria to recipient cells has been confirmed in multiple studies [15,32,33,34]; however, such ability of ultrapure MSCs (RECs) is not known. In this study, we validated the efficiency of mitochondrial transfer from RECs through three possible mitochondrial transfer routes (TNTs, Cx43-GJCs, and EVs) using contact and noncontact systems (Figure 9A–C) and assessed the benefit of this transfer in restoring mitochondrial function in EtBr-induced mitochondria-deficient (ρ^0^) cells. We found that RECs could transfer mitochondria to ρ^0^ cells via TNTs or Cx43-GJCs in the contact system, but there were deviations in transfer efficiency under different routes. In the noncontact system, RECs not only transferred large numbers of mitochondria to ρ^0^ cells but also alleviated the oxidative stress caused by impaired mitochondrial function and restored the mtDNA content. We also evaluated the effect of REC-derived exosomes on mitochondrial transfer and verified its positive regulatory role in the mitochondrial transfer pathway. These findings provide new insights that should be useful in the study of diseases associated with dysfunctional mitochondria.

TNTs, consisting of F-actin, myosin, and microtubulin, are the most popular routes for mitochondrial transfer between MSCs and recipient cells. MSCs have been shown to transfer mitochondria via TNTs to neuronal cells [35], cardiomyocytes [9], and lung epithelial cells [13], among others. This is consistent with our finding that mitochondrial transfer occurs through TNTs structured between RECs and ρ^0^ cells and that RECs transfer more mitochondria to ρ^0^ cells than BMSCs do. To clarify the proportion of mitochondria transferred from RECs through TNTs, we inhibited TNT formation by inhibiting actin polymerization with cytochalasin D. Despite this treatment, RECs transferred a large number of mitochondria to ρ^0^ cells. These results suggest that TNTs might not be involved in mitochondrial transfer from RECs to a considerable extent. The reasons for this could be that nonmotile cells typically accumulate a large number of microfilament bundles (stress fibers) [36] whose main constituent is F-actin and that the formation of stress fibers is usually accompanied by morphological changes in cells (cytoplasmic expansion) [37]. In contrast, RECs are characterized by smaller cells that are highly migratory and proliferative. In a previous study, we also found that the expression of F-actin in RECs is lower than in BMSCs [22]. F-actin, a major constituent of TNTs, confers antiflexing properties on the outward growth and protrusion length of TNTs [38] and crosslinking mitochondria to allow transport along the cytoskeletal structure of TNTs [39]. Thus, TNT-mediated mitochondrial transfer might not be predominant in RECs owing to their inherent characteristics. Moreover, it has also been reported that Cx43-mediated GJCs play a positive role in mitochondrial transfer by stabilizing the attachment of MSCs to recipient cells, thereby facilitating the formation of TNTs and EVs [18]. When GAP26 (a gap junction blocker) was added, we found a significant decrease in the rate of mitochondrial transfer from RECs compared with that from BMSCs. These results suggest a potential role of Cx43 in mitochondrial transfer from RECs; however, a synergistic effect of other mechanisms could not be completely excluded. Since Li et al. [40] published their data in 2002 on the discovery of mitochondrial localization of Cx43 in human umbilical vein endothelial cells, numerous studies have successively reported that mitochondria contain abundant Cx43 [41,42,43]. Correspondingly, we found an increase in Cx43 protein content in ρ^0^ cells after coculture, possibly because REC/BMSC donated mitochondria to ρ^0^ cells. Notably, we found a decrease in the Cx43 protein content in ρ^0^ cells when GAP26, a peptide corresponding to residues 63–75 of connexin 43, was added, which is a gap junction blocker. Related studies also suggest that GAP26 may directly inhibit connexin hemichannels and that gap junctions are subsequently inhibited [44]. Among them, gap junctions are an essential pathway for mitochondrial translocation, and their inhibition leads to a decrease in the rate of mitochondrial translocation, which affects the abundance of Cx43 protein.

EVs are membrane-bound particles composed of lipid bilayers (including MVs, EXOs, and apoptotic vesicles), which are essential carriers in intercellular communication involved in various diseases, as well as in biological events, such as immune responses and inflammation [45,46]. Furthermore, functional mitochondrial transfer mediated by EVs of MSC origin has been shown to promote anti-inflammation and OXPHOS in macrophages from lipopolysaccharide-induced lung injury mice models [47,48]. However, the relative involvement of EVs in REC-mediated mitochondrial transfer remains unclear. We examined the effect of EVs on the rate of mitochondrial transfer from RECs using the noncontact system, excluding the effect of material exchange from direct cell-to-cell contact, and MV formation was inhibited using Dynasore, thereby inhibiting mitochondrial delivery and reception. Without Dynasore treatment, RECs transferred large amounts of mitochondria to ρ^0^ cells compared to BMSCs. Moreover, we found that RECs transferred mitochondrial clusters into cocultured ρ^0^ cells in the form of MVs (Video S1). The MMP and OCR of ρ^0^ cells cocultured with RECs were significantly increased in the noncontact system. After Dynasore treatment, we found that RECs barely transferred mitochondria to ρ^0^ cells compared with BMSCs. Therefore, a possible role of MVs in the mitochondrial transfer from RECs is proposed, but that of EXO could not be excluded.

Furthermore, we found that RECs could restore the mtDNA content in ρ^0^ cells to a greater extent than BMSCs could (Figure 4H). Exosomes, owing to their specific structure, can deliver various substances (e.g., mtDNA, RNA, and proteins) from donor cells to recipient cells [49]. Previous studies have demonstrated that exogenous mtDNA can be transferred in vivo via EXOs and has functional consequences in OXPHOS-dependent breast cancer [50]. For this reason, we further analyzed the effect of REC- and BMSC-derived EXO on mitochondrial transfer. While identifying EXO, we found that REC-EXO and MSC-EXO were positive for exosomal markers, including CD9, CD81, and TSG101. In addition, TEM showed that both REC-EXO and MSC-EXO exhibited a typical exosomal spherical bilayer structure with a diameter range of 30–150 nm. The above results indicated the successful isolation of EXOs [51]. We added EXOs from different sources to ρ^0^ cells cocultured with RECs or BMSCs and found that the mitochondrial transfer rate was significantly increased, in contrast to the significant decrease in the rate of mitochondrial from RECs when EXO release was inhibited and compared with the rate of mitochondrial transfer from BMSCs. This illustrates the critical share of EXOs in mitochondrial transfer from RECs. In addition, EV-mediated mitochondrial transfer can increase the mitochondrial bioenergetics in the blood-brain barrier and neuronal endothelial cells [52]. We found that REC-derived EXOs increased the mtDNA levels in ρ^0^ cells and significantly restored the mitochondrial function of ρ^0^ cells. Moreover, REC-derived EXOs significantly increased the expression of Cx43 in ρ^0^ cells. The Cx43-GJC-mediated mitochondrial transfer pathway is also an essential pathway in MSCs [18]. These results point to a combined role of EXO and Cx43-GJCs pathways. Clarifying the possible involvement of different mitochondrial transfer pathways and the efficiency of different donor cells is essential for developing REC-based therapies, which would provide a foundation to enhance the mitochondrial transfer rate further and help in unraveling the underlying mechanisms.

Regenerative medicine using stem cells to treat mitochondria-related diseases has recently attracted the attention of scientists. Exogenous mitochondria derived from MSCs can be transferred to recipient cells and improve related functions through various pathways [33,53]. However, these BMSCs have limitations. The collection of adult BMSCs is highly invasive for the donor and carries a high risk of infection, allogeneic BMSCs are impure and have undifferentiated contaminating cells, and autologous BMSCs are costly [21,24,54]. Thus, highly homogeneous and safe BMSCs are critical to the study. In this study, we show the advantages of using RECs, which are ultra-purified and show very low batch-to-batch variation. RECs exhibit better characteristics in terms of cell proliferation, cell size uniformity, and expression of cell surface antigens [22,26]. Furthermore, we found that REC donated more mitochondria to ρ^0^ cells regardless of the coculture system and outperformed BMSCs in restoring the mtDNA content, partial mitochondrial function, and OCR. These findings suggest that REC mitochondrial transfer might be an effective strategy for treating diseases associated with mitochondrial disorders.

This study has several limitations. First, although we found the restoration of mitochondrial function in ρ^0^ cells after REC mitochondrial transfer, we did not explore the compatibility and retention times of such exogenous mitochondria in recipient cells. The transferred mitochondria or mtDNA may be eliminated by intracellular biological mechanisms. Second, although we elaborated on different mitochondrial transfer pathways, we did not perform an in-depth investigation of the synergism among the different pathways. Third, we did not confirm the pathophysiological correlation in vivo.

## 4. Materials and Methods

### 4.1. Cell Culture

Frozen, passage 5, RECs were purchased from PuREC Co., Ltd., Izumo, Japan. Vials of frozen, passage 2 or 4, BMSCs were purchased from Lonza (Basel, Switzerland). We prepared three different clones each of RECs and BMSCs. RECs and BMSCs were cultured in a complete culture medium, DMEM/Ham’s F-12 medium (FUJIFILM Wako Pure Chemical Corporation, Osaka, Japan) containing 15% HyClone fetal bovine serum (FBS; Cytiva, Tokyo, Japan), 2 mM L-glutamine, 100 U/mL each of penicillin and streptomycin, and 10 ng/mL basic fibroblast growth factor (bFGF; Abcam; Cambridge, UK) in a humidified atmosphere (5% CO_2_, at 37 °C) according to the manufacturer’s instructions. Cell culture medium was replaced every 2–3 days, and BMSCs from passage 4 were used (6 passages in total as for RECs).

Frozen A549 cells (adenocarcinoma human alveolar basal epithelial cell line) were obtained from RIKEN CELL BANK, Ibaraki, Japan. A549 cells were cultured in a complete culture medium, RPMI 1640 (FUJIFILM Wako Pure Chemical Corporation), supplemented with 2 mM L-glutamine, 100 U/mL each of penicillin and streptomycin, and 10% fetal bovine serum (FBS; HyClone, USA), in a humidified atmosphere (5% CO_2_, at 37 °C) according to the manufacturer’s instructions. Rho-null (ρ^0^) cells were established from A549 cells as previously described [55] through long-term treatment with 50 ng/mL ethidium bromide (EtBr; Sigma-Aldrich, St. Louis, MO, USA), 110 μg/mL sodium pyruvate solution (FUJIFILM Wako Pure Chemical Corporation), and 50 μg/mL uridine (Sigma-Aldrich) in RPMI 1640 (FUJIFILM Wako Pure Chemical Corporation, Tokyo, Japan), supplemented with 10% HyClone fetal bovine serum (FBS; Cytiva) for more than 20 passages.

### 4.2. Transfer of Mitochondria from RECs and BMSCs Using the Direct-Contact System

Prior to coculturing, BMSCs and RECs were labeled with MitoTracker Green FM (200 nM, Invitrogen, Carlsbad, CA, USA) for 20 min at 37 °C. The ρ^0^ cells were labeled with Vybrant™ DiD Cell-Labeling Solution (DiD) (5 μL/mL, Invitrogen) for 20 min at 37 °C. Cells were cocultured in RPMI 1640 medium (FUJIFILM Wako Pure Chemical Corporation), supplemented with 2 mM L-glutamine, 100 U/mL each of penicillin and streptomycin, and 10% FBS for 24 h. Cell viability was determined using a 4% trypan blue exclusion assay and light microscopy and was usually greater than 95%. Specific dye transfer was determined by gating DiD events, removing doublets (forward scatter width vs. height), and measuring fluorescence in the fluorescein channel. The mitochondrial relative transfer rate calculation method was as follows. CytoFLEX (BECKMAN COULTER, Brea, CA, USA) was used to sort DiD-labeled ρ^0^ cells (red), and MitoTracker Green-labeled BMSCs/RECs and intercellular mitochondrial transfer rates were calculated for the Q2 phase (double-positive) distribution as a percentage of total ρ^0^ A549 cells, normalized to the data. The data were analyzed using the FlowJo^TM^ Software Version 10 (BD, Franklin Lakes, NJ, USA). For mitochondrial transfer pathway inhibition experiments, we separately added the inhibitory compound (cytochalasin D, 350 nM; GAP26, 200µM) to the culture medium simultaneously with RECs or BMSCs during the assay.

### 4.3. Transfer of Mitochondria from RECs and BMSCs Using the Noncontact System

RECs and BMSCs were collected and placed onto a 3 μm cell culture insert (Corning, NY, USA) after mitochondria were labeled with MitoTracker Green FM (200 nM, Invitrogen). The ρ^0^ cells were labeled with DiD (Invitrogen), seeded in a 24-well plate, and cocultured with BMSCs or RECs. For inhibiting cellular endocytosis and, thereby, mitochondrial transfer, we added the inhibitory compound (Dynasore, 80 µM) to the medium at the same time as RECs or BMSCs.

### 4.4. Fluorescence Microscopy

To observe the distribution of mitochondria in RECs or BMSCs in direct coculture, mitochondria were stained using MitoTracker Deep Red (200 nM, Invitrogen) at 37 °C for 20 min before the coculture. Subsequently, after coculture with ρ^0^ cells for 4 h, the cells were fixed with 4% paraformaldehyde (PFA), permeabilized with 0.1% Triton X-100 in PBS, and blocked with 1% bovine serum albumin (BSA, Invitrogen) for 1 h. Thereafter, 1X green fluorescent phalloidin conjugate working solution (CytoPainter F-actin Staining Kit; Abcam) was added to the fixed cells, and the cells were stained for microtubules/TNTs for 60 min at 24 °C. Nuclei were stained with 1 μg/mL Hoechst 33342 (Invitrogen) for 20 min at room temperature. Fluorescent labeling was visualized using a BZ-X710 microscope (KEYENCE, Osaka, Japan).

To detect the expression of connexin 43, cells were stained with MitoTracker Deep Red (200 nM, Invitrogen) at 37 °C for 20 min prior to the coculture. After coculture with ρ^0^ cells for 4 h, cells were fixed with 4% PFA, permeabilized with 0.1% Triton X-100 in PBS, and blocked with 1% BSA for 1 h to block nonspecific binding. The cells were then incubated overnight at 4 °C with the primary anti-connexin 43 antibodies (1:1000, Lot: 83649S, Cell Signaling Technology, Danvers, MA, USA) prepared in PBS containing 1% BSA and 0.3% Triton X-100. Thereafter, the cells were washed three times with PBS (5 min for each wash) to remove excess antibodies and incubated with Alexa-Fluor^®^ 488-conjugated secondary antibody (1:500, Lot: 2110562, Invitrogen) for 1 h at room temperature. After three further washes of 5 min each, the cells were stained with 1 μg/mL Hoechst 33342 (Invitrogen) for 15 min at room temperature. After washing twice with PBS, the cells were analyzed using a fluorescence microscope (KEYENCE).

### 4.5. Western Blotting (WB)

For WB, proteins were extracted from cells and exosomes using RIPA buffer (FUJIFILM Wako Pure Chemical Corporation), supplemented with Halt^TM^ protease and phosphatase inhibitors (Thermo Fisher Scientific, Waltham, MA, USA). The cells were completely lysed by sonication with 40 pulses at high power. Finally, the lysates were centrifuged at 1500× *g* for 10 min at 4 °C to remove debris. The protein samples were quantified using the QubitTM protein assay kit (Invitrogen). Thirty micrograms of protein was electrophoresed on 12.5% SDS-polyacrylamide gel and transferred to cellulose nitrate membranes (ADVATEC^®^, Tokyo, Japan). The samples were blocked with 5% skimmed milk for 2 h at room temperature. After being washed three times using TBST buffer (Sigma-Aldrich), the samples were incubated overnight with a primary antibody (connexin 43 (E7N2R) XP^®^ rabbit mAb (1:1000 dilution in TBST, Lot: 83649S, Cell Signaling Technology); CD9 (D8O1A) rabbit mAb (1:1000 dilution in TBST, Lot: 13174S, Cell Signaling Technology); CD81 (D3N2D) rabbit mAb (1:1000 dilution in TBST, Lot: 56039S, Cell Signaling Technology); TSG101 (E6V1X) rabbit mAb (1:1000 dilution in TBST, Lot: 72312S, Cell Signaling Technology); GAPDH (1:3000 dilution in TBST, Lot: GR3272510-6, Abcam) at 4 °C. The blots were then washed three times with TBST, and incubated with the secondary antibody (Goat anti-rabbit IgG H&L (HRP), 1:4000 dilution in TBST, Lot: GR3455016-2, Abcam) for 2 h at room temperature. The blots were visualized using AmershamTM ECL Western Blotting Detection Reagent (Cytiva).

### 4.6. Isolation and Characterization of Exosomes

We used the ExoQuick-TC^TM^ exosome kit (System Biosciences, Palo Alto, CA, USA) to isolate EXO according to the manufacturer’s instructions. Briefly, RECs and BMSCs were cultured in a conventional FBS medium for a certain period of time. At 70% cell confluence, the original FBS-containing medium was removed and replaced with a fresh culture containing exosome-free serum (FBS, exosome-depleted; Capricorn Scientific GmbH, Ebsdorfergrund, Germany), and the culture was continued for 48 h. When the cell confluence reached approximately 90%, the culture supernatant was collected for EXO extraction. Residual cells and cell debris, including large EVs (apoptotic vesicles and MVs), were removed by centrifugation at 3000× *g* for 15 min at 4 °C. Ten milliliters of the supernatant was mixed with 2 mL of ExoQuick-TC precipitation buffer and incubated overnight at 4 °C. The supernatant was then removed by centrifugation at 1500× *g* for 30 min at 4 °C. After another round of centrifugation at 1500× *g* for 5 min at 4 °C, the supernatant was completely removed. The EXO particles were dissolved in RIPA buffer for use in WB. The morphology of EXOs was characterized using transmission electron microscopy (TEM). Nanoparticle tracking analysis was performed with a NanoSight NS300 instrument to analyze the particle size distribution and nanoparticle concentration of EXO. Specific EXO markers (CD9, CD81, and TSG101) were used as positive controls, and RECs/BMSCs treated with the EXO inhibitor GW4869 were used as negative controls.

### 4.7. Analysis of Mitochondrial DNA

QIAamp DNA Micro Kit (Qiagen, Germantown, MD, USA) was used to extract total DNA from each group of ρ^0^ cells. Total DNA concentration was quantified by measuring A260 and A280 using a NanoDrop spectrophotometer. Amplification of mtDNA was determined using PCR and corrected by measuring nDNA, as described previously [56].

ISOGEN-LS reagent (Nippon Gene, Tokyo, Japan) was used to extract total RNA from each group of ρ^0^ cells. Total RNA (2 μg) was reverse-transcribed to cDNA using PrimeScript II 1st Strand cDNA Synthesis Kit (TaKaRa, Shiga, Japan), following the manufacturer’s instructions. Real-time qPCR was performed in an Eco™ Real-Time PCR System (Illumina, San Diego, CA, USA), using KAPATM SYBR FAST qPCR Mastermix (KAPA Biosystems Inc., Woburn, MA, USA). The threshold cycle number (Ct) value was used to measure the relative expression of the input, and the difference in Ct values was used to quantify the expression of the target gene relative to the ACTB (housekeeping gene). The results are represented as the relative expression of the target gene calculated using 2^−ΔΔCt^, where ΔCt = Ct (target gene) − Ct (ACTB gene). The data were finally normalized based on the REC-treated ρ^0^ group. The expression of cytochrome oxidase-1 (*COX1*) and hypervariant and region 2 (*HVR2*) was used to verify the loss in mtDNA. The primers used to amplify the selected human genes are listed in Table 1.

### 4.8. Transmission Electron Microscopy

Observation of ρ^0^ cells by transmission electron microscopy was performed. The ρ^0^ cells were collected from the noncontact coculture (24 h) system. The collected cells were cultured in 35 mm plates. After 24 h, the cells were prefixed in 2.5% glutaraldehyde solution (FUJIFILM Wako Pure Chemical Corporation), 2% PFA, and 0.1 M phosphate buffer (FUJIFILM Wako Pure Chemical Corporation) for 2 h at room temperature, washed three times with PBS, and dehydrated in sequentially increasing concentrations of ethanol for 10 min. Then the cells were placed in epoxy resin, using propylene oxide for cell infiltration, and hardened overnight at 60 °C. The cells were then mounted on a copper mesh grid and stained with uranyl acetate. Morphological features of ρ^0^ cells were observed using TEM (Topcon EM-002B, Tokyo, Japan).

Exosomes were observed using TEM. EXO samples from REC or BMSC medium were stained to assess the uranyl acetate-negative status. A 400-mesh carbon-coated grid was loaded with 10 μL of freshly separated EXO samples. After 5 min at room temperature, additional 10 μL drops of 2% uranyl acetate were applied to the grid, incubated for 1 min, and the grid edges were dried using filter paper. TEM was used to examine the EXO morphology (Topcon EM-002B, Tokyo, Japan).

### 4.9. Measurement of Mitochondrial Membrane Potential

After coculture in a noncontact system, MMP was measured for different groups using the JC-1 MitoMP assay kit (Dojindo, Kumamoto, Japan), indicating potential-dependent accumulation of mitochondria in each group. Briefly, cells (8 × 10^4^ cells/mL) were incubated with 1 mmol/L JC-1 reagent for 50 min at 37 °C. The supernatant was removed, cells were washed twice with HBSS, and an imaging buffer solution was added to visualize the cells under a fluorescence microscope. The ratio of mitochondrial JC-1 aggregates to monomers was considered to be representative of the MMP of cells. The MMP of ρ^0^ cells was detected based on the ratio of red/green fluorescence intensity using a GloMax^®^ Discover Microplate Reader (Promega, Madison, WI, USA).

### 4.10. Measurement of ROS Levels

ROS production was detected with mitochondrial superoxide indicator MitoSOX probe according to the manufacturer’s protocol (Life Technologies, Carlsbad, CA, USA). After 12 h of coculture in a noncontact system, the cells were seeded in a 24-well plate and incubated with 5 μM MitoSOX working solution for 10 min at 37 °C, protected from light. After three gentle washes with a warm buffer (HBSS/Ca/Mg, Gibco, Paisley, UK), cells were subjected to confocal imaging (BZ-X710 microscope, KEYENCE). Fluorescence intensity was measured using CytoFLEX (BECKMAN COULTER).

### 4.11. Seahorse Analysis of Mitochondrial Function

Using a Seahorse XF HS mini analyzer (Seahorse Bioscience, Agilent, Santa Clara, CA, USA), we measured OCR, an indicator of mitochondrial respiration, according to the manufacturer’s instructions. Cells in each group were trypsinized and seeded into XF cell culture microplates (Seahorse Bioscience) at a density of 1.5 × 10^4^ cells/well one day before the assay. The following day, the culture medium was changed to XF Assay Medium (Seahorse Bioscience, Agilent) containing 10 mM glucose, 1 mM sodium pyruvate, and 2 mM L-glutamine, and transferred to a non-CO_2_ incubator for 1 h. The mitochondrial stress test was performed using Seahorse XF Cell Mito Stress Test Kit (Seahorse Bioscience, Agilent). First, 1.5 μM/well oligomycin (Olig) was used to quantify coupled and uncoupled respiration. Maximum respiration was measured using 1 μM/well FCCP. Spare capacity was evaluated using a combination of antimycin A and Rotenone (AA/Rot; 0.5 μM/well). Each plotted value for real-time mitochondrial respiration was represented as a percentage of the baseline OCR. OCR values were normalized to the total number of cells per well.

### 4.12. Statistical Analysis

Data analysis was performed using the GraphPad Prism 9 program (GraphPad Software Inc., San Diego, CA, USA). Data are expressed as mean ± SD. An unpaired 2-tailed Student’s *t*-test or one-way ANOVA with Tukey’s post hoc analysis was performed to compare the differences between two or more groups. Unless otherwise stated, values of * *p* < 0.05, ** *p* < 0.01, and *** *p* < 0.001 were considered significant.

## 5. Conclusions

We have highlighted the differences in mitochondrial transfer through different pathways (TNTs, Cx43-CJCs, and EVs), demonstrating the superior functional recovery of mitochondria-deficient cells using mitochondrial transfer from RECs. We demonstrate the feasibility of treating diseases associated with mitochondrial dysfunction using ultrapure, efficient, and homogeneous RECs.

## Figures and Tables

**Figure 1 ijms-24-10294-f001:**
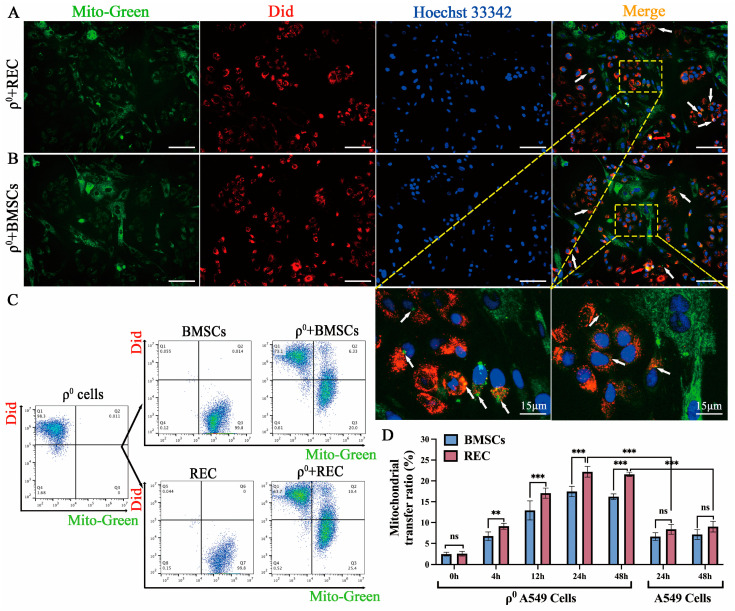
ρ^0^ cells receive mitochondria from REC or BMSCs. (**A**) Representative images of RECs (green fluorescence) with ρ^0^ cells (red fluorescence) in the direct-contact system. (**B**) Representative images of BMSCs (green fluorescence) with ρ^0^ cells (red fluorescence) in the direct-contact system. White arrows: Mitochondria transferred from RECs/BMSCs into ρ^0^ cells. Red arrows: transferred mitochondrial clusters. Scale bars, 100 μm. (**C**) Flow cytometry analysis of MitoGreen-stained RECs/BMSCs, DiD-stained ρ^0^ cells, and representative distribution under coculture. (**D**) Time course of mitochondrial transfer ratios between RECs/BMSCs and ρ^0^ cells at the Q2 phase, and between RECs/BMSCs and A549 cells (*n* = 3). Data represent the mean ± standard deviation of three independent experiments. Ns, not significant. ** *p* < 0.01, *** *p* < 0.001.

**Figure 2 ijms-24-10294-f002:**
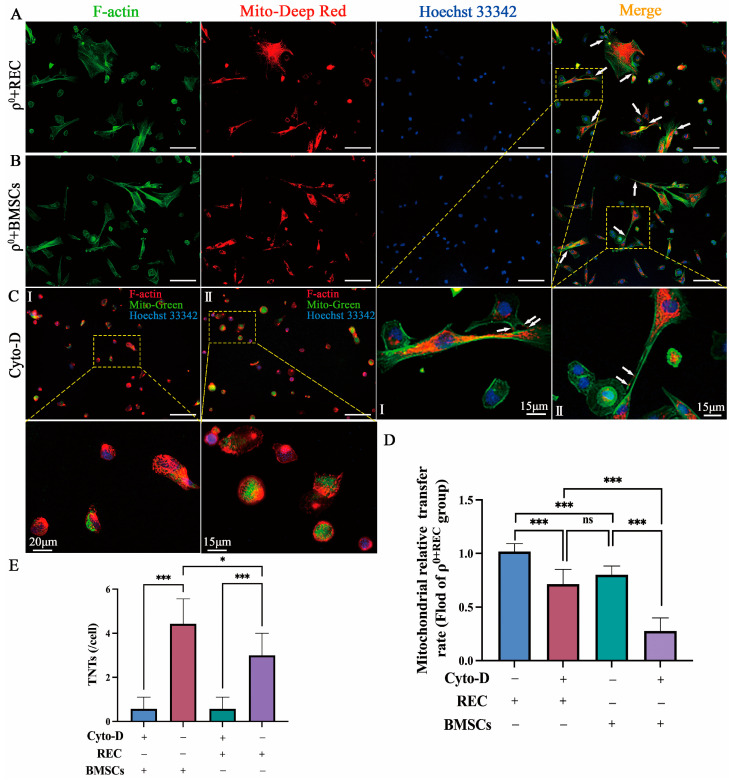
RECs/BMSCs donate mitochondria to ρ^0^ cells via tunneling nanotubes (TNTs). (**A**) Representative F-actin immunofluorescence images of RECs cocultured with ρ^0^ cells. (**B**) Representative F-actin immunofluorescence images of BMSCs cocultured with ρ^0^ cells. Mito-Red labeled RECs or BMSCs, and F-actin (green) and Hoechst33342 (blue) labeled all the cells. White arrows: mitochondria of RECs or BMSCs transferred into microtubules. Scale bars, 100 µm. (**C**) Representative immunofluorescence images of RECs (I) or BMSCs (II) cocultured with ρ^0^ cells after the addition of Cyto-D (a potent inhibitor of actin polymerization). Mito-Green labeled RECs or BMSCs, Hoechst33342 (blue) labeled ρ^0^ cells, and F-actin (red) labeled all the cells. Scale bars, 100 µm. (**D**) Relative rates of mitochondrial transfer from RECs and BMSCs to ρ^0^ cells with and without Cyto-D treatment (*n* = 3). (**E**) Number of TNTs between REC/BMSCs and ρ^0^ cells with or without Cyto-D treatment (*n* = 3). Cyto-D: Cytochalasin D. Data represent the mean ± standard deviation of three independent experiments. Ns, not significant. * *p* < 0.05, *** *p* < 0.001.

**Figure 3 ijms-24-10294-f003:**
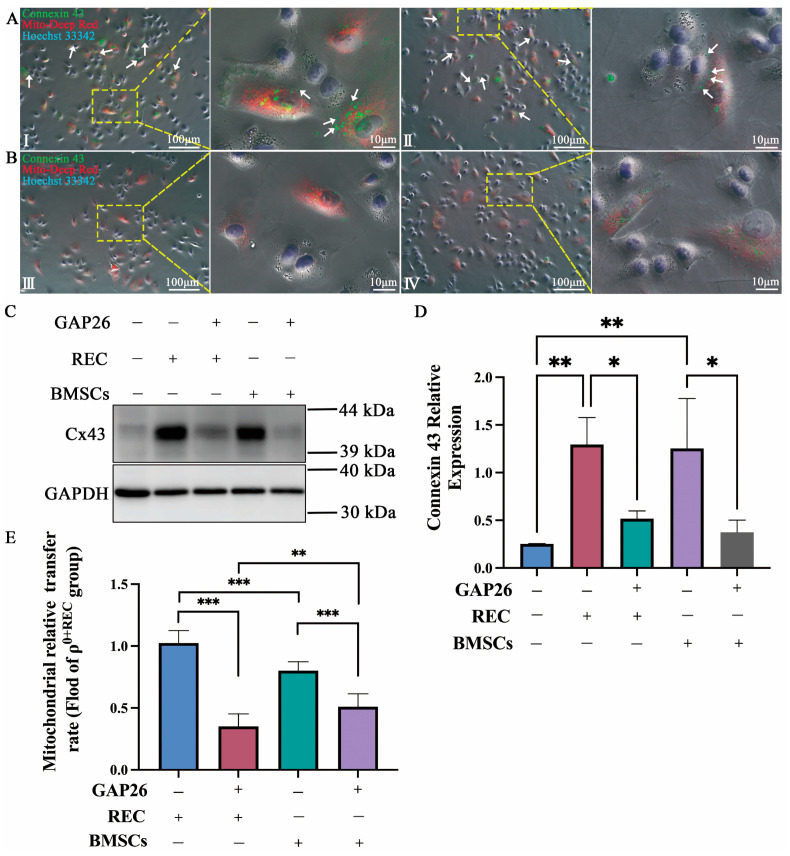
REC transfer mitochondria via Cx43-regulated gap junction channels. (**A**) Representative immunofluorescence images of RECs (I) or BMSCs (II) cocultured with ρ^0^ cells. Scale bar 100 µm. (**B**) Representative immunofluorescence images of RECs (III) or BMSCs (IV) cocultured with ρ^0^ cells after the addition of GAP26 (a gap junction blocker). Scale bar, 100 µm. Mito-Red labeled RECs or BMSCs, Hoechst33342 (blue) labeled ρ^0^ cells, and Cx43 (green) labeled all the cells. White arrows: High expression of Cx43 protein at REC or MSC junctions with ρ^0^ cells. (**C**) Western blotting analysis of Cx43 protein expression in coculture groups treated with 200 µM GAP26 or without GAP26, respectively. Representative blots were cropped from different experiments (*n* = 3). (**D**) Quantification of the Cx43/GAPDH ratio (*n* = 3). (**E**) Relative rates of mitochondrial transfer from RECs and BMSCs to ρ^0^ cells with or without GAP26 treatment (*n* = 3). Data represent the mean ± standard deviation of three independent experiments. * *p* < 0.05, ** *p* < 0.01, *** *p* < 0.001.

**Figure 4 ijms-24-10294-f004:**
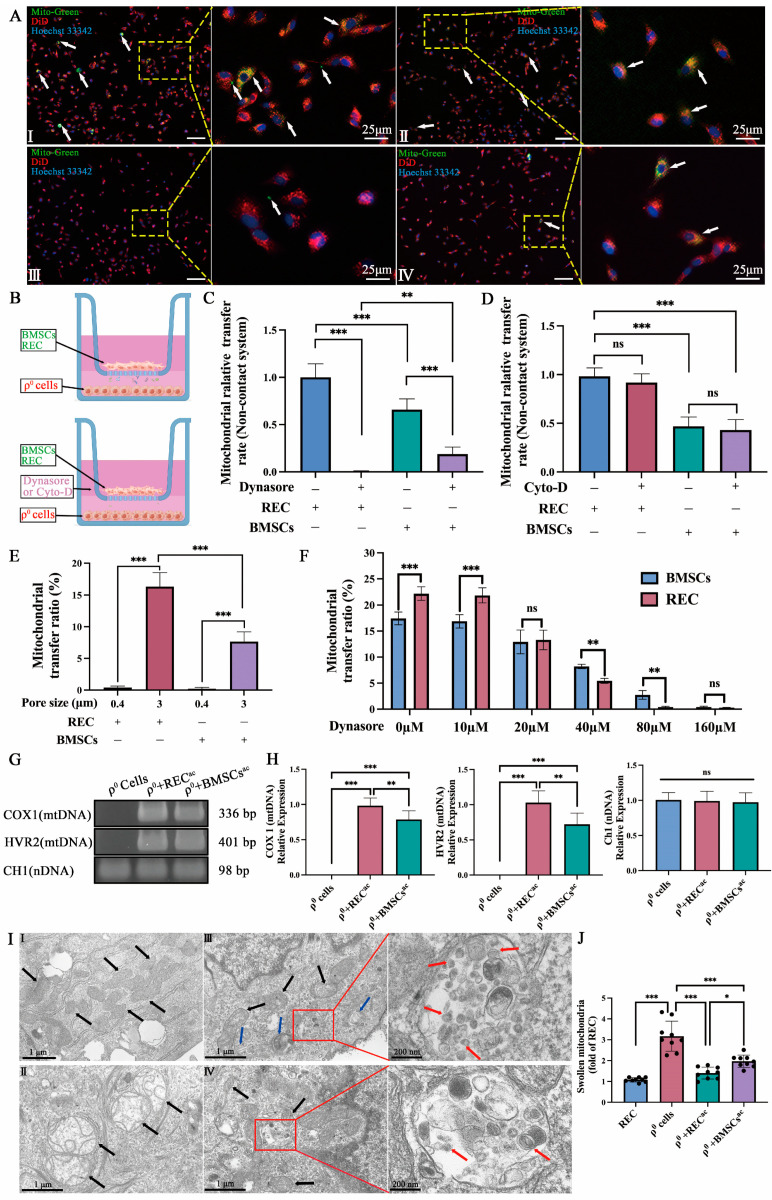
Mitochondrial transfer in the noncontact system. (**A**) Fluorescence images of ρ^0^ cells under coculture with RECs (I) and BMSCs (II), respectively, in a 3 μm pore size cell culture insert; ρ^0^ cells cocultured with RECs (III) and BMSCs (IV), respectively, in a 3 μm pore size cell culture insert with Dynasore. Mito-Green labeled RECs or BMSCs, DiD (red), and Hoechst33342 (blue) labeled ρ^0^ cells. White arrows: mitochondria from RECs/BMSCs received by ρ^0^ cells. Scale bar, 150 µm. (**B**) Schematic diagram of the noncontact system (with or without Dynasore/Cyto-D, an inhibitor of cellular endocytosis/inhibitor of TNTs). (**C**) Relative rates of mitochondrial transfer from RECs and BMSCs to ρ^0^ cells with and without Dynasore treatment (*n* = 3). (**D**) Relative rates of mitochondrial transfer from RECs and BMSCs to ρ^0^ cells with and without Cyto-D treatment (*n* = 3). (**E**) Rates of mitochondrial transfer from RECs and BMSCs in different coculture systems (0.4 μm and 3 μm pore size cell culture insert) (*n* = 3). (**F**) Rates of mitochondrial transfer from RECs and BMSCs at different Dynasore concentrations (*n* = 3). (**G**) Total DNA was isolated from ρ^0^ cells, ρ^0^ cells cocultured with RECs in the noncontact system (ρ^0^ + REC^ac^), and ρ^0^ cells cocultured with BMSCs in the noncontact system (ρ^0^ + BMSCs^ac^). PCR was performed to detect mitochondrial DNA (Cytochrome c oxidase I (*COX1*) and hypervariant region 2 (*HVR2*) and nuclear DNA (chromosome 1 segment (*CH1*). Notably, no mtDNA was detected in ρ^0^ cells. (**H**) RT-PCR quantification of the relative expression of *COX1*, *HVR2*, and *CH1* in ρ^0^ cells, ρ^0^ + REC^ac^, and ρ^0^ + BMSCs^ac^ (*n* = 3). (**I**) Representative TEM images showing the mitochondrial morphology of RECs (I), ρ^0^ cells (II), ρ^0^ cells cocultured with RECs for 24 h (III), and ρ^0^ cells cocultured with BMSCs for 24 h (IV). Black arrows: typical mitochondria of each group of cells; blue arrows: mitochondria in microvesicles; red arrows: apparently exosomes. (**J**) Quantification of mitochondrial swelling in different groups of cells (*n* = 3). Cyto-D: Cytochalasin D. ac: apical chamber: represents cells cultured in the apical chamber. Data represent the mean ± standard deviation of three independent experiments. ns, not significant. * *p* < 0.05, ** *p* < 0.01, *** *p* < 0.001.

**Figure 5 ijms-24-10294-f005:**
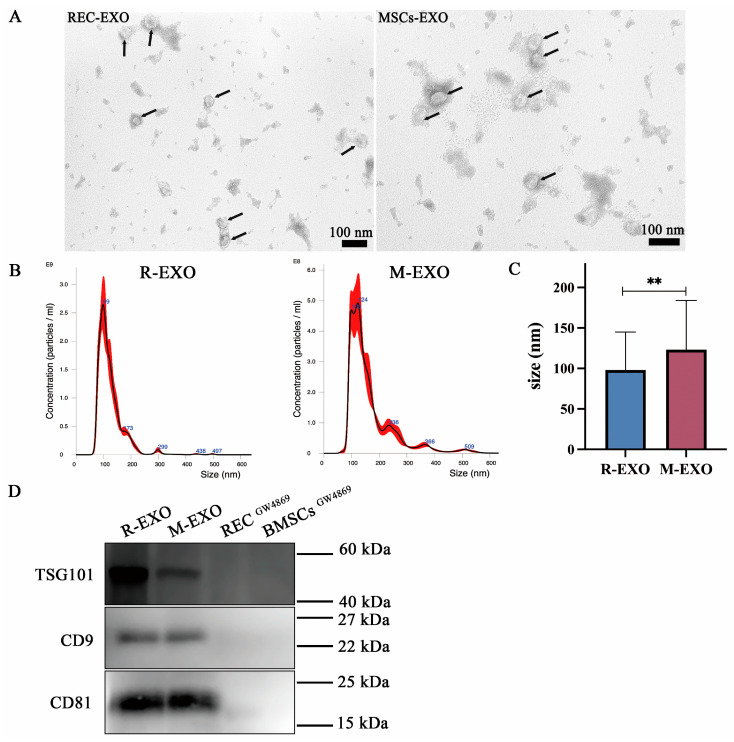
Identification of exosomes isolated from RECs and BMSCs. (**A**) Representative TEM images showing round or oval shapes and varied sizes of R-EXO and M-EXO with bilayer membrane structures. Black arrows: exosomes. (**B**) NanoSight was used to assess the particle size distribution of R-EXO and M-EXO. Error bars indicate ±1 standard error of the mean. (**C**) Quantitative analysis of particle diameter size (*n* = 3). Data represent the mean ± standard deviation of three independent experiments. ** *p* < 0.01. (**D**) Western blotting for detection of the exosomal markers (TSG101, CD9, and CD81) in R-EXO and M-EXO. R-EXO: REC-derived exosomes. M-EXO: BMSC-derived exosomes. GW4869 is an inhibitor of exosome biogenesis/release.

**Figure 6 ijms-24-10294-f006:**
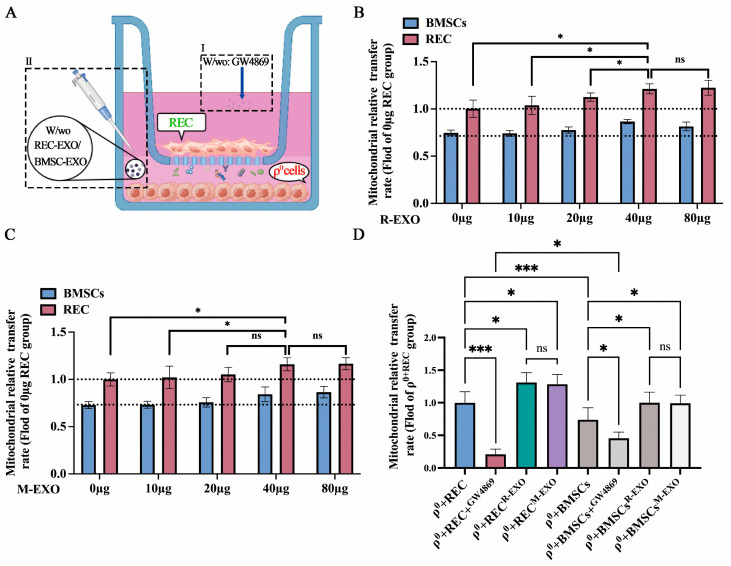
Augmentation of the mitochondrial transfer rate by exosomes. (**A**) A schematic diagram of the coculture system shown with the upper layer of the chamber seeded with RECs or BMSCs and the lower layer seeded with ρ^0^ cells and cocultured with or without GW4869 (I). When ρ^0^ cells were cocultured with RECs or BMSCs, they were treated with or without R-EXO or M-EXO (40 μg per protein) (II). (**B**) Rates of mitochondrial transfer from RECs and BMSCs were detected using flow cytometry at different R-EXO concentrations (*n* = 3). (**C**) Rates of mitochondrial transfer from RECs and BMSCs were determined using flow cytometry at different M-EXO concentrations (*n* = 3). (**D**) Flow cytometry measurement of the mitochondrial transfer ratio between Mito green-labeled (green) RECs or BMSCs and DiD-labeled ρ^0^ cells (red) (*n* = 3). R-EXO: REC-derived exosomes. M-EXO: BMSC-derived exosomes. GW4869 is an inhibitor of exosome biogenesis/release. Values represent the mean ± standard deviation of data from three independent experiments. ns, not significant. * *p* < 0.05, *** *p* < 0.001.

**Figure 7 ijms-24-10294-f007:**
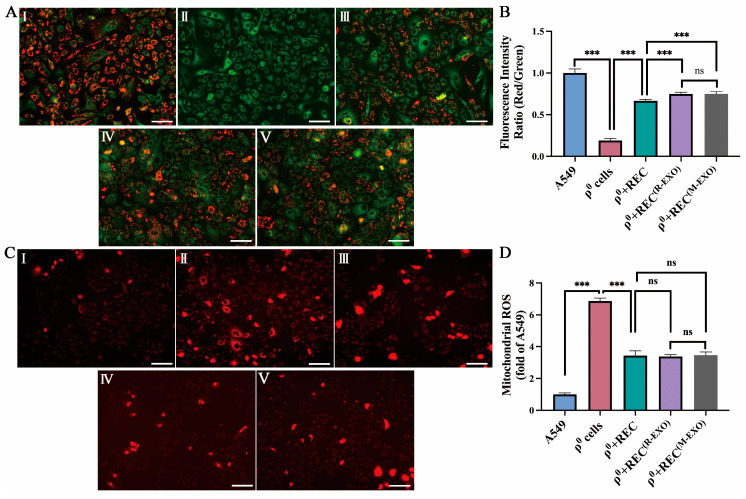
Exosomes elevate mitochondrial function and mtDNA content in ρ^0^ cells. (**A**) Representative fluorescence imaging of MMP: I, A549 cells; II, ρ^0^ cells; III, ρ^0^ cells cocultured with RECs (ρ^0^ + REC); IV, ρ^0^ cells cocultured with RECs were supplemented with exosomes derived from RECs (ρ^0^ + REC^(R-EXO)^); V, ρ^0^ cells cocultured with RECs were supplemented with exosomes derived from BMSCs (ρ^0^ + REC^(M-EXO)^). Scale bar, 100 µm. (**B**) Quantitative analysis of the red/green (JC-1) fluorescence ratio of MMP in each group of cells (*n* = 3). (**C**) Representative fluorescence imaging of Mito-Sox: I, A549 cells; II, ρ^0^ cells; III, ρ^0^ + REC; IV, ρ^0^ + REC^(R-EXO)^; V, ρ^0^ + REC^(M-EXO)^. Scale bar, 100 µm. (**D**) Quantitative analysis of mitochondrial ROS levels in each group of cells (*n* = 3). (**E**) PCR was performed to detect *COX1*, *HVR2*, and *CH1*. (**F**) RT-PCR quantification of the relative expression of *COX1*, *HVR2*, and *CH1* in ρ^0^ cells, ρ^0^ + REC, ρ^0^ + REC^(R-EXO)^ and ρ^0^ + REC^(M-EXO)^ (*n* = 3). (**G**) WB analysis of Cx43 protein expression in ρ^0^ cells; ρ^0^ cells were treated with REC-derived exosomes (ρ^0 + R-EXO^) or ρ^0^ cells were treated with BMSC-derived exosomes (ρ^0 + M-EXO^). Representative blots were cropped from different experiments (*n* = 3). (**H**) Quantification of the Cx43/GAPDH ratio (*n* = 3). Values represent the mean ± standard deviation of data from three independent experiments. ns, not significant. * *p* < 0.05, *** *p* < 0.001.

**Figure 8 ijms-24-10294-f008:**
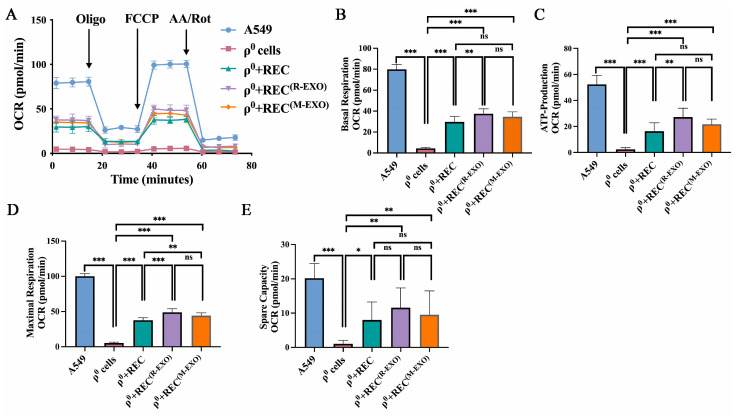
Attenuation of oxidative stress-induced mitochondrial respiratory dysfunction by R-EXO. (**A**) Mitochondrial OCR of A549, ρ^0^ cells, ρ^0^ + REC, ρ^0^ + REC^(R-EXO)^, and ρ^0^ + REC^(M-EXO)^ was measured using the Mito Stress kit (*n* = 3). The order of oligomycin, FCCP, and AA/Rot injections is given. Basal respiration (**B**), ATP production (**C**), maximal respiration (**D**), and spare capacity (**E**) were analyzed for each group of cells (*n* = 3). Values represent the mean ± standard deviation of data from three independent experiments. ns, not significant. * *p* < 0.05, ** *p* < 0.01, *** *p* < 0.001.

**Figure 9 ijms-24-10294-f009:**
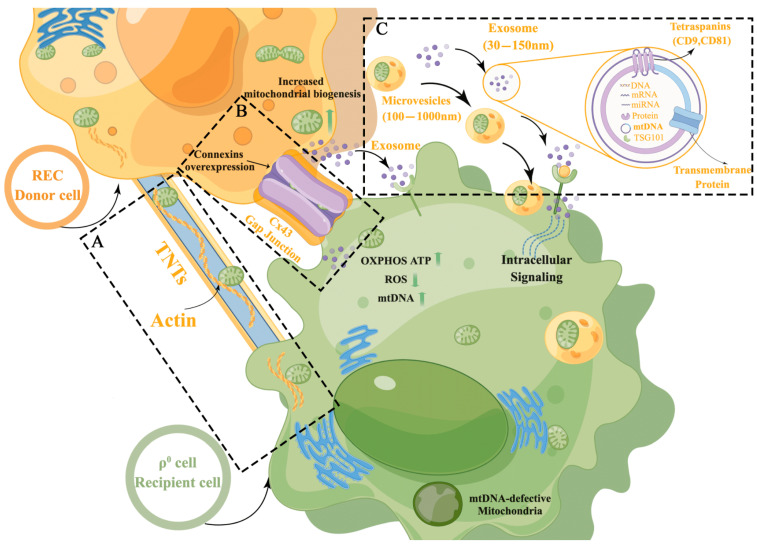
Schematic diagram of the mechanism of partial mitochondrial transfer from RECs to ρ^0^ cells. (**A**) TNTs provide one of the common mitochondrial transfer routes, generally connecting the cytoplasm of cells via F-actin, and RECs can donate mitochondria to mitochondrial-deficient (ρ^0^) cells via this intercellular linkage structure. (**B**) Cx43 gap junction channels (GJCs) are also involved in the transcellular transfer of mitochondria. RECs can donate mitochondria to ρ^0^ cells via GJCs. (**C**) Extracellular vesicles (EVs) can transfer mitochondria to recipient cells via endocytosis, where REC-derived exosomes have a positive regulatory role in the mitochondrial transfer process. The figure was prepared using Figdraw.

**Table 1 ijms-24-10294-t001:** Primer pairs to confirm the establishment of ρ^0^ cells.

Gene Name	Primer Pairs	Product Size (bp)
*COX1* for real-time PCR	Forward 5′-GCT ACC ATA ATC ATC GCT ATC-3′	155
	Reverse 5′-GCT AAT ACA ATG CCA GTC AG-3′	
*HVR2* for real-time PCR	Forward 5′-CTA TGT CGC AGT ATC TGT CT-3′	86
	Reverse 5′-AGT AAG TAT GTT CGC CTG TA-3′	
*COX1* for PCR	Forward 5′-ACA CGA GCA TAT TTC ACC TCC G-3′	336
	Reverse 5′-GGA TTT TGG CGT AGG TTT GGT C-3′	
*HVR2* for PCR	Forward 5′-CTC ACG GGA GCT CTC CAT GC-3′	401
	Reverse 5′-CTG TTA AAA GTG CAT ACC GCC A-3′	
*CH1*	Forward 5′-GGC TCT GTG AGG GAT ATA AAG ACA-3′	98
	Reverse 5′-CAA ACC ACC CGA GCA ACT AAT CT-3′	
*ACTB*	Forward 5′-TGG CAC CCA GCA CAA TGA A-3′	186
	Reverse 5′-CTA AGT CAT AGT CCG CCT AGA AGC A-3′	

## Data Availability

The datasets used and/or analyzed during the current study are available from the corresponding author upon reasonable request.

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
