# Peer review of "Extracellular Vesicles and Cx43-Gap Junction Channels Are the Main Routes for Mitochondrial Transfer from Ultra-Purified Mesenchymal Stem Cells, RECs"

_ijms, 2023, doi:10.3390/ijms241210294_

Round 1
Reviewer 1 Report
In this research article, Jiahao Yang et al., investigated the transfer of mitochondria from rapidly expanding clones and mesenchymal stem cells to a human mitochondria-deficient cell line. The authors investigate the transfer via Tunneling Nanotubes, Gap junction internalization and trafficking, as well as via exosomes.
There are certain issues, which are being highlighted below:
Major:
In general, all experiments should be performed for at least n=3 and specified for each analysis.
Introduction:
Line 67: Both references describe different ways of mitochondrial transfer (TNT vs. internalization). Mitochondrial transfer via GJ internalization is shown in Ref. 17, but there is no proof of OXPHOS restoring, so the citing context should be changed.
Results:
Section 2.1: The explanation of ρ0 as mitochondria-deficient A549 cells comes first in line 112 and should be moved to line 97.
Figure 1: The mitos are hard to see in the shown low mags, I recommend a higher magnification as it is shown in Figure 2. An explanation for the white arrow in the figure legend is missed. Please mention n numbers of experiments.
Section 2.2: Line 125: I can´t see how TNTs connect communication, TNTs connection enable communication, the phrase should be refined.
Figure 2: The high mags of TNTs with labelled mitos are quite pretty, but in (C) the shown cells do hardly connect with each other, not to mention a mitochondrial transfer is observed. Why show cells distinct morphology to co-cultured cells in (A) and (B). In my opinion, the shown images are not convincing, and should be changed by more meaningful higher mags. Perhaps, phase contrast images could be useful to show the cell borders independent of mitochondrial staining. (D) How was the mitochondrial relative transfer rate calculated? Please add a short explanation in the materials and method section. Please check the y axis label for spelling mistakes (this also applies for the following figures). Please mention n numbers of experiments (D,E).
Section 2.3: Line 162: What is meant by linker protein? Cx43 is a component of gap junctions. Please refine.
Figure 3: Check the labelling of images in (C) (“CAP26”) and the figure legend. (D) I wonder about the low protein expression level in ρ0 cells. Is this a known observation for mitochondria-deficient cells? A549 wild type cells typically express Cx43. At least three marker bands should be marked for each protein. Please discuss reasons for the decreased Cx43 expression after GAP26 inhibition. The decrease in protein expression is not mandatory for GJ inhibition. (E,F) Please check the y axis label for spelling mistakes.
Section 2.4: (B) I recommend a higher magnification, as double staining is hard to see. (D,E,F) Abbreviations for cells should be consistent, therefor “MSC” should be corrected to “BMSC”. Please check the y axis label for spelling mistakes.
Section 2.5: Line 233 and 243, please explain abbreviations “MMP” and “OCR”.
Figure 5: (A) Why are those cells not homolog in their morphology? Please unify the abbreviation for your cells (see section above). (J) Explanation of black arrows is missing.
Line 301ff: Is the observation of an increased mitochondrial transfer after adding additional EXOs to ρ0 cells not a logical effect as more EXOs with mitos results in an increase of mitochondrial transfer alone through the high amount of transferable mitos?
Section 2.6: Figure 6: At least three marker bands should be marked for each protein.
Section 2.7, Figure 7: (B) Check y axis label.
Section 2.8: Are those experiments performed in the non-contact system?
Figure 8: (F) How is the relative expression calculated? (G) At least three marker bands should be marked for each protein. (H) Check y axis label.
Figure 9: What is meant by the Basal mitochondrial OCR?
Discussion:
Figure 10 is slipped to the materials & methods section. The figure should be moved to the discussion.
Materials and Methods:
Section 4.2: Line 514: Please correct typ “CAP26”.
Section 4.4: Please indicate the product numbers of primary and secondary antibodies.
Section 4.5: Where antibody incubation performed in TBST or in skim milk in TBST?
Section 4.7: Please check the primer pairs as listed primer pairs for COX1 and HVR2 are far to big for use in real-time PCR. According to this section, only A549 wild type cells and ρ0 cells were used for RT-PCR. Data A549 wild type cells are not shown in Figure 4, why are they mentioned? Please correct this section according to the shown data. How was the relative expression calculated?
Reviewer 2 Report
This paper compares routes of mitochondrial transfer from two types of mesenchymal cells to an energetically deficient cell line in culture. Such a comparison is of relatively high interest because of recent attention that intercellular mitochondrial transfer has received, along with proposals of mechanisms mediating such transfer.
In addition to the specific concerns detailed below, I have serious reservations about this study:
1) A very recent paper by this group compared efficacy of mitochondrial transfer from MSCs and RECs to EthBr selected HeLa and A549 cell lines, with many figures showing in most cases similar results. And the potential routes of transfer were explicitly considered, and even illustrated in Fig 1 of that paper. Thus, novelty of higher efficacy of RECs in transfer to the compromised A547 cells is not novel, and also the hypothesized transfer mechanisms. Rather, the merits of the present paper rest on dissection of the route of transfer.
2) As such, relative contribution of each pathway must be quantitatively compared, which is not done well in the figures in this manuscript (whereas in Yang et al, 2023, such a comparison is presented in Fig 5 for both HeLa and A549 cell lines).. The terms “many”, “several” and “some” are not quantitatively meaningful. The phrase “more number” must be replaced throughout.
3) In both this paper and the one that has just appeared, authors declare no conflict of interest. Because this paper reports advantageous results with the REC cells, obtained from PuREC Ltd which appears to be affiliated with the same university as the authors (and potentially may involve them), these relationships must be disclosed in detail.
4) Fonts in figures are too small to be legible; micrographs are too small to see; figure labels must be checked carefully for misspellings.
5) Efficacy, selectivity and specificity of treatments used to interfere with transfer pathways are not well described
6) The energy-deficient cells were obtained by EthBr treatment of A547 cells, and may thus represent a subpopulation with other characteristics that differ from the parent. Comparisons should thus be interpreted cautiously.
Specific comments:
Fig 1. Images are too small to see details; suggest enlarge or provide higher resolution images in Supplement. Fig 1 C and D are very similar to results presented in the authors’ recent paper, although they differ in the thresholds chosen for separation of the transferred mitochondria (Quadrant I). Quantification also differs, in that Fig 2F of previous paper showed significant difference ar 4 & 8 hr coculture, whereas present Fig1D shows significant difference only at 12 hr. It is also curious that the difference in mitochondrial transfer is not seen in the parental A547 cells.
Fig 2 and TNTs (repeating findings more compellingly shown in their recent paper). In order to compare TNT-dependent and -independent mitochondrial transfer, the authors treated cells in contact with cytochalasin, with the assumption that it specifically inhibited TNT formation (and similarly with both REC and BMSC donors). However, criteria for identifying TNTs are not defined (and processes seem too thick in most cases to qualify as typical TNTs), and mitochondria relationship to microtubles is not clear. Moreover, the rather small difference in transfer of REC vs BMSC mitochondria at 12 hr in the absence of cytochalasin is in contrast to results shown in Fig 1 of this paper and Fig 2 of the previous one). Is cytochalasin specific for TNTs, or does it also affect GJs? . The image labels in Fig 2C are also too small and especially confusing since the colors are switched compared to what was used in the rest of the figure.
Fig 3 and GJs (slightly extending their recent publication). This is very confusing to me. L162, for example: Cx43 is not a linker protein, rather it forms the channels. The immunostaining in Fig 3 shows only cytosolic Cx43 staining, whereas Cx43 is expected to form plaques between cells. In the figure and elsewhere, the purported GJ inhibitor, GAP26, is misspelled as CAP26. It is unexpected that coculture itself would radically increase the amount of Cx43 and that the peptide would change its abundance. Fig 3 does not provide compelling immunophotographs that this peptide interferes with mitochondrial uptake from either donor cell type, which should be striking considering the histogram in F. How do the authors’ envisage Cx43-ediated transfer? If through GJ channels, there are numerous blockers available eg, fenamates, carbenoxolone, as in their recent paper); if through expression, antisense or other strategies might be applied. It is also noted that many of these findings regarding Cx43 were reported in the previous paper, iwht the exception of the GAP25 data.
Fig 4 and noncontact transfer. TNTs are tiny, as are other types of slender processes connected through gap junctions, and the reliance on 3 um filters for total interference of contact requires validation (in the previous paper, 0.5 um filters abolished the interaction). Use of the GTPase inhibitor Dynasore to block endocytosis may be justifiable, but endocytosis is a likely route for gap junction-mediated transfer and its effect on TNTs in this system need to be determined. It is curious that inhibition of mtRNA transfer by dynasore is not shown. Nothing is really visible for Fig 4B at the magnification shown. It would be better if zoomed images were provided like what was done for TNT visualization in Fig 2. • Complete blockage of mitochondria transfer is reported after Dynasore treatment (endocytosis inhibition) in the contact-less system with RECs but not BMSCs (Fig 4C). What would account for the remaining transfer if the cells are not making contact?
Fig 5 and restoration of mitochondrial function (which largely overlaps with Fig 8 in the previous Yang et al publication). The substantial improvement in energetics in the “noncontact” coculture is impressive and repeats what was previously shown by this group. Missing, however, is comparison to comparison with coculture in contact, in order to evaluate the extent to which contact plays a role. Also missing is validation that cells do not contact one another under these coculture conditions, as in Fig 4. •The TEM conclusions in Fig 5 are confusing, in particular the description of “structures similar to exosomes in extracellular vesicles.”
Fig 6. The EM images show profiles that are oval in many cases, but not necessarily spherical and not uniformly. What do the red bars represent on the hisograms? Are REC and BMSC EV sizes statistically different? The western blot markers used have also been reported for microvesicles and are not conclusive of exosome purity. An exosome inhibitor was used but they do not provide information on reduced number of vesicles following inhibition. The western was also not quantified and did not include a control.
Fig 7. The phospholipase inhibitor GW4869 shouid be defined on l297 as well as in the legend. How was the amount of EXO added (40 ug) decided upon? Was there dose-dependent effect? A rescue experiment would be informative in Fig 7 (GW4869 inhibition + isolated exosome application)
Fig 8. B, D, F show tiny differences between mitochondrial uptake and energetics after EXO compared to REC that could in principle be due to higher concentration of mitochondrial fragments or other components in the EXO…how is concentration justified? The increase in Cx43 is curious; is Cx43 detectable in the EXOs?
Fig 9. Again, effects of EXO are impressive, but claiming superiority to REC requires assumptions about concentration.
Use of a special character to denote the subline is strongly discouraged, as inconvenient for reference by others.
Covered in critique
Round 2
Reviewer 1 Report
The authors spent a lot of time in the correction. The high mags obviously contribute to the clarification.
However, my comments on the western blots were misleading. I recommend the article of Tie et al. (Tie, L., Xiao, H., Wu, Dl. et al. A brief guide to good practices in pharmacological experiments: Western blotting. Acta Pharmacol Sin 42, 1015–1017 (2021). https://doi.org/10.1038/s41401-020-00539-7): “The positions of molecular weight markers should be shown or marked on all the blot images. If the blots have been cropped horizontally, at least two neighboring marker positions (i.e., above and below the bands) should be indicated, as shown in Fig. 1c.” Therefor, the authors should insert the marker bands in Figures 3 and 6 accordingly. In addition, the images of the third experiment should be added to the file “original images”.
Reviewer 2 Report
This manuscript is greatly implorved through higher magnification images and rewriting, although significant concerns remain.
1) The authors misunderstood my previous commentabout quantitation, “More number” must be replaced through out by “higher number” or, simply “more”.
2) Potenitla COI is not adequately addressed
3) Magnification of some figures remains inadequate to see details
4) Previous review provided detailed comments regarding figures that overlap between this manuscript and the one recently published by this group. If data are duplicative, figures could be removed or provided as supplemental information.
L63. No, Cx43 does not transfer mitochondria (published claim is that it participates in such transfer)
Fig 1. Please check scale bars; insets indicate cells are 50 um diameter, very large, and “mitochondrial clusters in lower mag images seem also this large. How does one imagine that 50 um structures transfer through any of the nanoscale mechanisms proposed here?
L160 What does this mean? Cx43 is an essential component of GJCs . Is Cx43 the only/main gap junction protein in these cells?
Fig 3 and its legend. Western blot shows very large reduction in Cx43 level after GAP26, seemingly larger than the 75% difference in panel E. This is at odds with tiny SD atop the histogram bar. The explanation that this Cx43 is being transferred due to its presence in mitochondria would require assurance that the major pool of Cx43 within the cells is in mitochondria, which seems implausible. Cx43 staining must be shown at higher magnification (at least 20 x that presented here) to convincingly show that it is present at appositional regions; moreover, the argument that GAP26 is interfering with exchange requires that these GJ plaques would be absent following treatment, which is not shown.
L220. These are concentrations, not gradients
Ll 225-6. Rewrite …”…cyto-D did not affect…..(Cytochalasin D)”
Fig 4. Scale bars or magnification information is lacking for higher magnification images…check other figures as well.
L347 Change “related scholars”
Fig 7B-D. Although there is higher basal mito transfer in the new cell type than BMSCs, this could transfer could result from many factors, such as higher surface area, membrane binding characteristics, etc. The histograms in B and C seem to be overselling the data; although statistical comparison is lacking in B & C, differences shown in panel D with and without M/R-EXO are very similar for the two cell types.
Numerous grammatic corrections required. Mulitple uses of "More number" appears to have resulted form misunderstanding of my previous comment.
Round 3
Reviewer 2 Report
The authors have corrected all problematic figures and datasets, although new figure 3 does not adequately show Cx43 plaques. THe use of pastel colors does not provide high enough contrast and the lime green color should be replaced with a darker tint. Also, although figure legend indicates that arrows are black, l 171 states that they are white. width of arrowhead should be increased and that of the haft thinned to provide better idenfication of structures to which they point.
l174 states that GAP36 is "derived from", whereas the statement on l450 that it corresponds to the sequence is more accurate.
CX43 should be changed to Cx43 throughout.
English usage is adequate, though could be improved by editing
